# Smart Systems Implementation in UK Food Manufacturing Companies: A Sustainability Perspective

**Andrew Thomas [1],\***, **Claire Haven-Tang [1]**, **Richard Barton [1]**, **Rachel Mason-Jones [1]**, **Mark Francis [1] and Paul Byard [2],\***

[1] Cardiff School of Management, Cardiff Metropolitan University, Cardiff CF5 2YB, UK; chaven-tang@cardiffmet.ac.uk (C.H.-T.); bartonrichard2@sky.com (R.B.); rkmason-jones@cardiffmet.ac.uk (R.M.-J.); mfrancis@cardiffmet.ac.uk (M.F.)

[2] Engineering Employers Federation, Wales CF31 3WT, UK

\* Correspondence: ajthomas@cardiffmet.ac.uk (A.T.); pbyard@eef.org.uk (P.B.)

**Abstract:** The UK food industry faces significant challenges to remain sustainable. With major challenges, such as Brexit, on the horizon, companies can no longer rely on a low labour cost workforce to maintain low production costs and achieve economic sustainability. Smart Systems (SS) is being seen as an approach towards achieving significant improvements in both economic and environmental sustainability. However, there is little evidence to indicate whether UK food companies are prepared for the implementation of such systems. The purpose of this research is to explore the applicability of Smart Systems in UK food manufacturing companies, and to identify the key priority areas and improvement levers for the implementation of such systems. A triangulated primary research approach is adopted that includes a questionnaire, follow-up interviews, and visits to 32 food manufacturing companies in the UK. The questionnaire and interviews are guided by a unique measuring instrument that the authors developed that focusses upon SS technologies and systems. This paper makes an original contribution in that it is one of the few academic studies to explore the implementation of SS in the industry, and provides a new perspective on the key drivers and inhibitors of its implementation. The findings suggest that the current turbulence in the industry could be bringing food companies closer to the adoption of such systems; hence, it is a good time to define and develop the optimum SS implementation strategy.

**Keywords:** food manufacturing; digital hub; sustainability profile; smart systems; survey

## 1. Introduction

The UK's food sector is complex and highly dynamic in nature. The demands placed upon the manufacturing system through short-life products and raw materials, more demanding retailers and end users, and increased levels of legislation and regulation have resulted in organisations needing to respond on multiple levels and on a range of different issues in order to achieve economic and environmental sustainability [1]. In some cases, these pressures have resulted in the sector becoming increasingly isolated from other manufacturing sectors as they deal with their own specific problems [2]. The resulting problem of this isolation is that many food manufacturing companies are not necessarily aware of the advances in manufacturing technologies that are being made in, and the systems that are being developed and applied throughout, the wider manufacturing industry. This, in turn, can lead to the creation of an environment where the food manufacturing industry may be left behind when it comes to adopting and benefitting from new and advanced manufacturing technologies [3].

Isolation of the sector, and further isolation from individual problems and symptoms at a business unit level, threatens the economic sustainability of food manufacturing companies and the sector as a whole. Major retailers offer these food manufacturing companies the greatest potential for increased sales, job creation, and efficiency of production. However, this has to be reconciled with the demands of reduced profit margins and increased costs that are associated with higher volume requirements [4,5].

In order to cope with these business pressures, other manufacturing and production sectors have placed increasing focus upon the development and advancement of technology-driven manufacturing systems, such as Smart Factories, Smart Systems, and Industry 4.0 (I.E. 4.0). These systems are often known collectively as Smart Systems (SS). Recent years have seen step change improvements in terms of Smart Systems' capability, reduced cost of technology, and wider accessibility of the skills and knowledge required to implement them. Therefore, it is possible to articulate the current challenge within the UK food manufacturing industry in terms of two distinct objectives aimed at overcoming their isolation and aligning their businesses towards Smart Systems implementation. These objectives are:

1. Through the development and application of a SS/Sustainability profiling tool in 32 food manufacturing companies, to understand the current expertise and identify the technological priorities of the UK food manufacturing companies when considering the implementation of Smart Systems; and
2. To propose a conceptual system architecture for effective SS implementation.

Evidently, effective implementation is the key to success, and learning from experience in implementing other business improvement practices and paradigms shows that there is no single prescriptive implementation guide to fit every company. So, this paper takes an important early view of enablers of, and potential barriers to, success and presents them in the context of an implementation framework to be easily leveraged across both the UK and international food sector in order to minimize the learning curve costs and timescales.

## 2. Literature Review

UK food manufacturers are highly aware of the need to operate within visible supply chains. Smart Systems provide this essential link in that the technologies and systems enable an improved level of traceability right through the manufacturing chain, where machines are interconnected and, archiving data can be done automatically [6]. Alongside this, environmental tracking can be better achieved as well as monitoring energy usage so that optimising energy consumption profiles can be achieved. On the whole, the likely result of the adoption of SS in the food manufacturing sector will result in improved machine performance, optimised maintenance, and reduced costs [6–8]. This should then provide new opportunities for companies to win new customers and retain existing ones. It is also likely to create new revenue streams in the form of value-adding services and, allow for seamless connectivity with upstream and downstream supply chain partners [6].

The industrial trend towards the adoption of Smart Systems is based largely on the perceived positive benefits that cyber-connected, automated systems can bring to industry and meeting the sustainability agenda, such as improved efficiency, greater customisation, improved quality, reduced waste, and enhanced economic sustainability [6]. For instance, Bonilla et al. [9] link four different business scenarios (deployment, operation and technologies, integration, and compliance) with sustainable development goals. From these scenarios, their analysis resulted in a number of positive and negative sustainability impacts being identified when related to the basic production inputs and outputs flows (raw material, energy and information consumption, and product and waste disposal). Therefore, further work is required in the form of a more detailed literature analysis of how SS can meet the sustainability agenda. Section 2.1 develops this work.

## 2.1. Smart Systems: A Literature Review

Smart Systems (SS) can be defined as the employment of manufacturing and communication technologies to allow higher levels of interconnectivity, leading to greater communication between machines and decentralised/local processing of data [10]. SS embraces a wide range of technologies, including Radio Frequency Identification (RFID), Near Field Communication (NFC), Wi-Fi, Cellular, and Bluetooth, all linked to networks that normally use the Internet as a form of communication [9,10]. SS technologies offer many benefits that link to the key sustainability dimensions, including the ability to improve food traceability, reduce food waste, and increase efficiencies in the transport and handling of food products, and in turn contribute directly to addressing both economic and environmental sustainability challenges [11]. On a wider scale, the virtualization of supply chains using SS technologies enables companies to optimise supply chain operations and characterise the dynamic nature of operations [11]. Virtualisation also enhances the opportunity to apply innovations and improvements in supply chains, and to subsequently plan for and assess these innovations without affecting the manufacturing system. It also enables innovative thinking amongst staff and the promotion of the view of what new and innovative technologies can do to enhance productivity and product innovation [12] as well as addressing the economic sustainability challenges [13]. Today, the technology is highly reliable, relatively cheap, and based on international standards that promote easy communication between different devices' tags and systems [11]. A further and more detailed review of the literature on Smart Systems, the technologies, and its impact on the sustainability dimensions is shown in Table 1.

**Table 1.** An analysis of the literature on smart systems.

| Smart Systems Research Clusters | Smart Technologies and Systems | Sustainability Dimensions |
| --- | --- | --- |
| Time compression, time to market. | Three-dimensional (3D) Printing, simulation, virtual reality (VR), customer integration, virtualization [11,14,15] | Reduced development time and tooling cost [16]. |
| Sustainable Product Innovation | Intelligent product design systems [17,18] | Inter-functional collaboration, innovation-oriented learning, research and development (R&D) investment [17]. |
| Human Factors | Innovation, competency management [19,20] | Work practices, social dimensions, human rights, ergonomics, and safety [19]. |
| Knowledge Management | Intelligent Decision Making: predictive scheduling, fuzzy logic systems [21,22] | Organisational and deep-learning systems [23]. |
| Energy Systems | Energy-neutral technologies through Internet of Things (IoT) [24] | Waste reduction and energy monitoring [25]. |
| Enterprise Reconfiguration | Rapid supply chain reconfiguration through IoT & Cyber Physical Systems (CPS), Virtualization [11,26] | Value Mapping and information sharing tools [27]. |
| Collaborative Networks | Customer/supply chain connectivity [26] | Company/Knowledge base collaboration [28], e-Word of Mouth (e-WOM), and Digital marketing [29–32]. |
| Management Systems | Technology management, control, and monitoring [21,22] | |
| Digital Systems | Digital supply chains, data analytics, cyber physical systems [9,10,33,34] | Big data analytics on environmental impacts [35]. |

## 2.2. Smart Systems: An Analysis of the Literature

Through undertaking an analysis of academic articles that focus on Smart Systems and their connectivity to sustainability, it is possible to identify nine key smart system clusters that have emerged from the work and are shown in Table 1. The analysis has further identified the key SS technologies and systems as well as the connectivity between SS and the sustainability dimensions. This analysis suggests that SS technologies and systems are at an advanced stage of development, and the connection between the sustainability dimensions means that the move towards the employment of SS in industry is likely to impact greatly (and positively) on improving the sustainability of companies, especially in the economic and environmental sustainability dimensions. Furthermore, this literature analysis as well as supporting evidence, such as [8], suggests that the food industry is ideally placed to benefit

from adopting SS. The continuous demand to maintain and often reduce costs in the food industry means that companies have to continuously innovate and develop more efficient manufacturing systems as well as seek to innovate the product in order to maintain cost levels. SS is likely to be seen as a significant opportunity for companies to potentially stabilise productivity and improve output both in terms of cost reduction and quality consistency [8,36]. The greater flexibility offered by SS will enable a product volume mix to be achieved with greater levels of consistency and efficiency. In many cases, bespoke manufacturing can be achieved as well as the capacity to rapidly change to meet differing customer demands as a result of such technologies and systems [9].

So, if the food manufacturing industry is ideally placed to take advantage of SS, then why is the industry slow to pick up on the concept and implement such systems? The traditional barriers towards the implementation of Advanced Manufacturing Technologies in the past have focused upon the high cost of technology and the limited capability of the existing workforce to operate and develop the technologies [13]. However, with the emergence of relatively inexpensive internet-based technologies and systems, why are these barriers still relevant today? The research question for this study is, therefore: "*what are the current capabilities and priorities of the UK food manufacturing industry to meet the requirements of Smart Systems implementation?*".

As a result of the adoption of SS, food companies will need to focus on a different knowledge skillset, and will, therefore, need to recruit, upskill, and keep staff capable of maintaining these highly complicated business operations [37,38]. However, evidence suggests that UK Food companies may not be fully aware of the benefits that SS can bring about [18,39,40]. It, therefore, seems that the food industry in general lacks the knowledge and an understanding of the need to implement new and sometimes advanced technologies in its businesses [41,42].

In summary, the benefits that SS can bring are appreciated by industrialists and academics alike. Improved product traceability, including traceability in the food recall system [41], improved productivity throughput, shorter processing times, and improved consistency of product quality are all seen as positive elements of SS implementation [43]. The falling cost of technologies as well as the ubiquitous nature of internet connectivity combined with relatively powerful computing equipment raises the question as to whether the traditional impediments of technology cost and worker skills are still seen as major barriers, or whether these issues remain perceptions based on a previous era of manufacturing.

In order to further understand these issues that were identified from this literature analysis, the authors undertook a small-scale survey of 32 UK food manufacturing companies of various sizes with the aim of identifying the level of awareness of SS within their companies, and to also identify the dynamics of technology adoption. Using the SS clusters and technologies and systems that were identified in this literature analysis as the main guide to the development of the survey tool, the authors undertook the survey to obtain some baseline information on how industry leaders view SS adoption in their companies.

## 3. The Research Method and Survey Design

A triangulated research approach was employed, consisting of the following stages:

1. An analysis of secondary research obtained from academic sources.
2. A small-scale pilot survey of food manufacturing companies (Stage 1 research study).
3. Follow-up interviews with Managing Directors (MDs) and Managers from the small-scale pilot study (Stage 2 research study).

The Stage 1 research process required the development of an appropriate SS profiling tool that could be used to measure specific responses from the companies but also act as a point of reference for discussion on SS implementation. The authors developed a sustainability profiling tool that is primarily based on the work from the previously undertaken literature analysis and further literature on manufacturing challenges and SS systems [44–48]. The profiling tool is shown in Table 2 The tool

utilises the SS research clusters, SS technologies, and sustainability dimensions that were highlighted from the literature review and detailed in Table 1 of this paper to form the main body of the tool.

**Table 2.** The sustainability profiling input sheet.

| Smart Systems Sustainability Clusters | Smart Technology Areas | Average Current Level of Expertise | Average 2 Year Priority Score | Gap | Frequency (Current Expertise) | | | | |
|---|---|---|---|---|---|---|---|---|---|
| | | | | | 1 | 2 | 3 | 4 | 5 |
| Time Compression, Time to Market (Ec) | V1 Customer Integration with product development process | 4.3 | 4.75 | 0.5 | 0 | 1 | 2 | 15 | 14 |
| | V2 Application of time compression technologies | 3.85 | 4.5 | 0.7 | 0 | 1 | 11 | 12 | 8 |
| Sustainable Product Innovation (Ec) | V3 Robust New Product Development/Introduction (NPD/I) | 4.4 | 4.65 | 0.3 | 0 | 0 | 1 | 16 | 15 |
| | V4 Intelligent and Customised products | 3.95 | 4.45 | 0.5 | 0 | 2 | 8 | 12 | 10 |
| Human Factors (Ec/En) | V5 R & D Systems/Co-Innovation/creativity | 3.45 | 4.25 | 0.8 | 3 | 4 | 8 | 9 | 8 |
| | V6 Competency management | 3.1 | 4.75 | 1.7 | 5 | 6 | 7 | 7 | 7 |
| Knowledge Management (Ec/En) | V7 Organisational Learning systems | 1.9 | 4.75 | 2.9 | 14 | 10 | 5 | 3 | 0 |
| | V8 Intelligent decision-making systems | 4.15 | 4.75 | 0.6 | 0 | 0 | 8 | 12 | 12 |
| Energy Systems (En) | V9 Waste Reduction Systems | 4.3 | 4.85 | 0.6 | 0 | 0 | 3 | 17 | 12 |
| | V10 Energy neutral production systems | 3.6 | 5 | 1.4 | 3 | 2 | 8 | 11 | 8 |
| Enterprise Reconfiguration (Ec/En) | V11 Information-Sharing Systems | 2.55 | 4.4 | 1.9 | 8 | 9 | 7 | 5 | 3 |
| | V12 Rapid Supply Chain Reconfiguration | 3.8 | 4.25 | 0.5 | 0 | 2 | 11 | 11 | 8 |
| Collaborative Networks (Ec/En) | V13 Customer and Supply Chain Collaboration | 3.4 | 4.1 | 0.7 | 2 | 6 | 8 | 9 | 7 |
| | V14 Company/University Collaboration | 2.3 | 4.9 | 2.6 | 7 | 14 | 8 | 2 | 1 |
| Management Systems (Ec/En) | V15 Manufacturing Fitness | 4.05 | 4.4 | 0.4 | 0 | 0 | 9 | 13 | 10 |
| | V16 Technology Management Systems | 4.2 | 4.6 | 0.4 | 0 | 0 | 5 | 16 | 11 |
| Digital Systems (Ec/En) | V17 Digitally Connected Supply Chains | 1.6 | 4.85 | 3.3 | 16 | 13 | 2 | 1 | 0 |
| | V18 Data analytics and Production Analytics | 1.55 | 4.65 | 3.1 | 16 | 15 | 1 | 0 | 0 |

Note: Abbreviations: Ec, Economic Sustainability Driver; En, Environmental Sustainability Driver; Ec/En, both.

Companies were selected by the research team based on the following definition of a food manufacturing company: "being primarily concerned in converting raw ingredients and products into food products, and identified as mass production/high volume companies employing high-volume manufacturing systems and configurations" [40]. One hundred and thirty requests were issued electronically to food manufacturing companies, asking the MDs of each company to take part in the survey. Thirty-two companies responded and agreed to undertake the survey. Table 3 shows the companies and food sectors that responded to the survey, and the size of each company measured in terms of the number of employees. The companies that were involved in the Stage 2 study are marked in brackets.

**Table 3.** The companies and sectors that responded to the survey: Stage 1 Stage 2.

| Sectors | Companies per Sector | Employees 10–50 | Employees 50–150 | Employees 150–200 |
|---|---|---|---|---|
| Packaging and Logistics | 4 (3) | 1 (1) | | 3 (2) |
| General Drink | 2 (1) | | 2 (1) | |
| Wines, Beers, and Spirits | 5 (2) | 3 (1) | 2 (1) | |
| Ready meals and processed foods | 5 (4) | 2 (2) | 3 (2) | |
| Cheese and Dairy | 4 (3) | 2 (2) | 2 (1) | |
| Bread, Bakery, and Snacks | 10 (5) | 6 (4) | 3 (1) | 1 |
| Biscuits, cake, and chocolate | 2 (2) | 1 (1) | | 1 (1) |
| Totals | 32 (20) | 18 (11) | 10 (6) | 4 (3) |

During the profiling stage, each company was contacted, and a time arranged for a member of the research team to visit the company. The initial stage of the study involved a member of the research team meeting with the MD of each company to discuss the sustainability profiling. The profiling stage

involved a discussion about each strategic driver and an explanation of what each of the drivers and associated technologies meant in order to ensure that there was a common understanding about the meaning of each driver. The research member in discussion with the MD then completed the profiling exercise. This score was then validated by the researcher undertaking a detailed observational study of the systems and technologies employed within the company. A short moderation session followed the observation and interview with the MD to ensure that a consensus was achieved on each driver and dimension that was scored.

Scores were assigned to each strategic driver and associated indicative technology, which initially focused upon the current level of expertise the MD believed that their company had against the 18 technology/systems dimensions highlighted. The second stage of scoring required the MD to prioritise each dimension based on a two-year planning horizon (i.e., where they thought their company needed to be to meet the demands of their industry). This profiling allowed the team to determine the current state of operational excellence and also the strategic intent of each company in meeting the SS requirements. The gap between the current state and the aspirational level 2 years into the future provides the basis for discussion in Stage 2 of the research study.

Following the profiling exercise, the researchers moved to the Stage 2 research study. The Managing Directors and Senior Management of 20 companies from the original survey group agreed to be interviewed further through unstructured face-to-face interviews. The aim of these interviews was to discuss further the responses provided from the Stage 1 study and to understand the complex nature of the priority areas that were highlighted by the surveyed food manufacturing companies.

## 4. The Results of the Survey and Interviews

A synopsis of the Stage 1 sustainability profiling results is shown in this section of work. Table 3 shows an average score of the 32 food manufacturing companies on their assessment of their current technological expertise, and also their two-year technology priority score. Furthermore, the table also shows a frequency analysis that profiles the score each company provided against each technology area. This enabled the researchers to understand the relative level of expertise each company had in relation to the technology areas. Figure 1 focusses specifically upon the sample group's average current expertise profile in ranked order. Taking the top four criteria from this figure shows that the companies' new product development and introduction capabilities, along with their customer integration, waste reduction, and technology management expertise, were considered to be strong and well-developed. Where the companies scored less-well were in the lower four criteria, namely knowledge base collaboration, organizational learning, digital connectedness, and data analytics. Figure 1 also shows the average 2-year priority scores offered by the sample group of companies. The 2-year priority profile is a measure of what the companies considered to be the key technologies and systems that need to be in place in order for the companies to remain competitive over the medium-term strategic planning horizon. The figure shows that the top four priority areas to focus on are: energy-neutral production systems; competency management; digitally connected supply chains; and university/company collaboration. The four criteria of lower concern are: supply chain reconfiguration; customer and supplier collaboration; information sharing; and Research and Development and Innovation.

### 4.1. Analysis of Results (Stage 1 Study)

The findings of the Stage 1 survey are shown in Figure 1. The figure represents the current overall scores from all 32 food manufacturers as well as the scores split between the Small SMEs (Small and Medium Enterprises) (18 companies) and the Medium SME/Large companies (14 companies). The figure also shows the 2-year priority profile for the 32 companies. The overall findings of the current scores were not particularly surprising. Food manufacturing companies have traditionally developed strong NPD/I systems that involve close collaboration with customers. Likewise, the management of their current manufacturing systems and technologies as well as the development of robust waste reduction systems is well-known. Likewise, areas that receive less attention, such as collaboration

with knowledge bases and a lack of understanding of digital connectivity and data analytics, is also well-known within the industry. Therefore, the common issues that can be found within the wider food manufacturing industry are accurately reflected within this smaller sample group.

An analysis of the 2-year technology priorities showed that companies were very aspirational in implementing and developing state-of-the-art technologies and systems. In particular, the focus on reducing energy consumption and moving towards energy-neutral manufacturing systems is interesting, since companies felt that their waste reduction strategies were relatively well-advanced, but company energy-reduction strategies needed further work and development. Of further interest was the identification of the priority to have 'digitally connected supply chains'. Although seen as a strategic priority, the companies did not see themselves having the current expertise (or knew where to access the expertise) in order to move towards this priority area. This issue links strongly with the disparity seen between the current overall lack of development in the areas of competency management, knowledge management, and University/company collaboration. However, the companies did see that these areas were critical for meeting their future strategic intent as there was a clear lack of understanding amongst the surveyed companies that, in order to move to the adoption of Smart Systems, there needed to be a greater development of staff and further collaboration with Smart Systems experts that are very likely to exist outside the food industry. The external drivers, such as Brexit, outweighed the potential barriers and internal issues, such as the costs of training and equipment, as they saw the threat of significant external change as being greater than the internal resistance that had previously been seen. Further analysis of the data identified that the Small SMEs (10–50 employees) performed better on the whole in the deployment of internet and smart systems technologies, and were better aligned to meeting the social, environmental, and economic sustainability goals. Although their technologies and systems lacked the sophistication of the larger companies, the application of internet and cyber physical systems pertaining to their own production operations were better developed. This particular issue was further developed in the Stage 2 research study. A particularly well-developed area amongst the Small SME companies is the development of excellent supply chain collaboration practices between customer and supplier that are delivered through internet technologies (internet and social media platforms).

Through the development of closer collaboration within the supply chain, small SMEs benefited from greater opportunities to develop more customised products and services through the co-creativity of new products and innovative solutions to particular production issues, thus creating a virtuous circle for these companies. A particular strength of the medium-to-large companies was their ability to manage their technologies and to operate lean production systems as well as utilizing time compression technologies, such as automated production systems and the simulation of new production layouts for a new product's introduction. However, whilst these technologies are utilized and well-developed, their overall connectivity to Cyber Physical Systems (CPS), which provide the basis for Smart Systems, was missing in all companies surveyed.

Therefore, two distinct patterns emerge from this study that emphasise the difference in attitudes between Small SMEs, Medium SMEs, and larger companies. Smaller SMEs use less sophisticated technology but utilize their systems to better effect, linking their technologies to both the customer and the supplier in more of a traditional Smart Systems approach, whereas medium-sized SMEs and larger companies employ more sophisticated technologies, but they lack the interconnectivity and CPS technologies to turn their technology into Smart Systems. The next section of the paper will focus on Stage 2 of the research programme, which involved undertaking further and more in-depth interviews with the MDs and senior managers of 20 companies that participated in the Stage 1 research programme. The aim was to attempt to understand further the issues around SS development within the companies and to highlight the drivers of, and barriers to, SS implementation.

*4.2. Analysis of Results (Stage 2 Study)*

The responses that were obtained from the companies can be grouped into two strategic themes, namely company strategy and manufacturing strategy.

*Company Strategy*: The findings of the Stage 1 phase of the study showed that the companies saw that investment in SS technologies and systems was critical to their survival. The driver for implementing such systems over the next 2 years was, however, driven by concerns over the rise in labour costs that was driven in turn by major political changes around Brexit and the actual and potential loss of highly skilled European workers. Most companies commented that they had lost, on average, 30% of their skilled workforce due to the threat of Brexit, and had previously gone through the pain of training and developing local workers, but had largely failed to retain that workforce. The potential quality problems emanating from the need to employ new staff was also seen as a potential future concern. Therefore, with the potential need to employ new and inexperienced staff in a post-Brexit era, company directors now saw the switch to SS and its associated technologies as being more realistic considering that a significant change in company strategy was needed to respond to the potential political change.

A secondary finding from the interviews highlighted an important issue around future worker recruitment and retention in that companies in general envisaged that the adoption of SS would enhance the image of the industry towards being one that was more sophisticated in nature, more environmentally friendly, and a more exciting and challenging industry to work in. It was envisaged that the 'knock-on' effect to this image change would be that more talented, technologically focused workers would be drawn to the industry, thus reducing the concerns over attracting talent into the industry.

*Manufacturing Strategy*: In order to remain competitive and, therefore, economically sustainable, the primary focus of development within the medium-sized SMEs and larger companies was on the continual improvement of manufacturing performance, whereas most Small SMEs focused upon innovation and new product development as a means towards maintaining a competitive advantage. Therefore, as can be expected, the type of SS and the associated technologies differ considerably (i.e., the need for highly automated and connected SS for larger manufacturing-focused food manufacturers compared to the more internet-connected, social-media-oriented smaller SMEs, where new products and ideas are identified through closer connectivity with their consumers).

The drive for automated manufacturing SS within the larger companies was driven by their focus on the continuous improvement of manufacturing capacity and capability. This was primarily down to the issue that the companies surveyed were mainly food processors and had little responsibility for product development. Most MDs saw this as a major concern for future sustainability, and believed that having responsibility for the development of new products would enable the company to have longer-term viability.

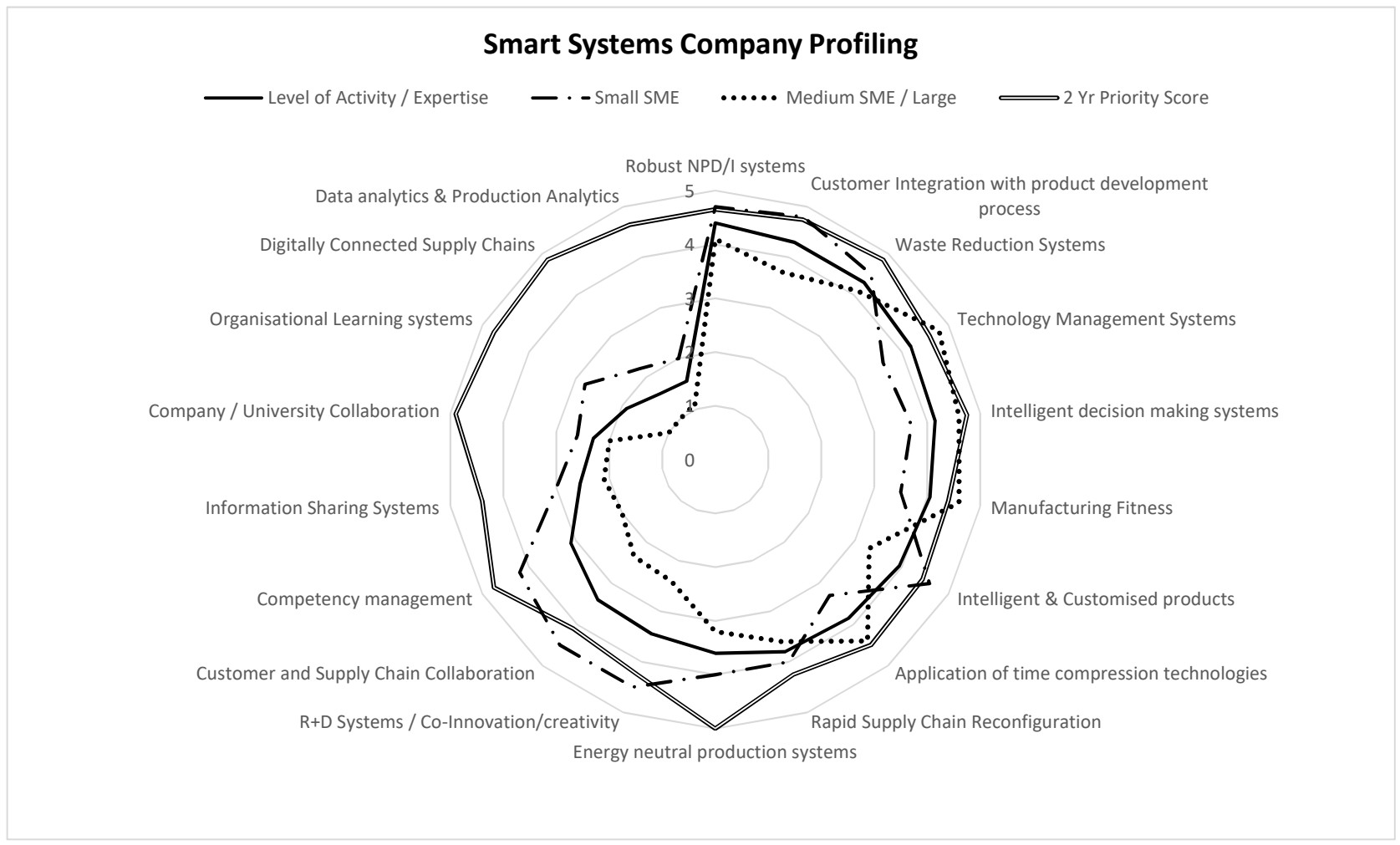

**Figure 1.** The analysis of current and future profiles in ranked order.

In discussions with the MDs of the smaller SMEs who had a greater focus on smaller production volumes but greater involvement in the design, development, and introduction of new food products, it was clear that their respective manufacturing strategies did not involve expanding their businesses to cater for any significant increase in manufacturing demand, and that they preferred to collaborate with larger manufacturers and to outsource responsibility for production if demand so dictated. Therefore, smaller SMEs were more likely to seek collaborative solutions with other companies. However, this was not the case with larger food manufacturers, who sought to deal with production issues, such as new product development, by themselves rather than de-risk the NPD process through supply chain collaboration with smaller but more expert companies. Therefore, one of the barriers to larger companies failing to invest in NPD systems and, thus, to remaining manufacturing-only plants was the perceived costs of investment and the risk of failure.

## 5. Discussion

Company respondents identified the continuing pressure on their companies to continually innovate and also to reduce production costs and increase production yield in order to remain economically sustainable. As a result of these pressures, most saw the need to acquire a greater level of automation [49]. Most of the responses from medium-to-large companies involved moving towards 'lights-out manufacture' and 24:7 manufacturing, and this would rely heavily upon automated systems and technologies. Many large-scale manufacturing facilities already operate partially automated systems. However, the shift towards web-based, integrated, and automated systems that will ensure an increase in productive yield, and product quality that is consistent and repeatable, has not yet been made.

The SS technologies and systems that were seen as being crucial for implementation in company facilities are highlighted as follows. *Big data and knowledge-based automation*: in collecting, analysing, and making sense of a wide range of production data and semantic data from multimedia/social media [50], allowing companies to understand customer preferences and personalise products. *Smart Systems*: the immediate application of Smart Technologies and systems is to enable businesses to optimize production and also resource management and energy minimization throughout the supply chain [10,51]. *Advanced and autonomous systems*: moving routine food-manufacturing operations, such as food preparation and cleaning activities, to autonomous and near-autonomous activities through the use of computer vision, sensors, including GPS, and remote-control algorithms [52]. *Cloud computing Computerised food manufacturing execution systems*: working in real-time to enable the control of multiple elements, including enhancing productivity, supply chain management, collaboration, resource and material planning, and customer relationship management [49,53,54].

*New management approaches for Smart Systems*: The demands for ensuring the security and reliability of food availability requires serious changes in the way food manufacturing functions are managed. Improving distribution, increasing productivity, and reducing waste though a range of initiatives, such as enhancing food supply, better network planning of outlets and distribution to maximise efficiency and improve resilience, multiple uses of crops/waste streams, and novel processes to minimise water and energy requirements, are all key issues requiring new management paradigms to effectively manage the complexity of such systems [55]. These issues can be further enhanced through managing Smart Systems through Cloud management, Big Data Analytics, and intelligent decision-making systems [56]. Allied to these issues is the need for *New skills and Knowledge Bases*: future knowledge generation and leadership that will enable the development of "digital thinking" so that companies manage the process in a new way and allow for quicker and more accurate decision-making [57,58].

Whist many of these technologies and systems will be focused upon the large food manufacturing companies and secondary production providers (such as packaging, logistics, and warehousing), more elementary yet critical technologies and systems are required by the small food suppliers. The respondents from the small food companies that were surveyed identified that developing a

sound knowledge of digital marketing and e-Word of Mouth (e-WoM) [29–32] is in great demand in order to ensure that smaller companies achieve greater visibility with a wider range of customers and more immediate feedback from clients in order to remain at the forefront of the product development process [59]. Allied to this issue is the enhancement of a company's use of Social Media systems to include correct website development with enhanced capabilities for order-making, payments, and special product requests. Key to the enhancement of SME capabilities is the need to establish strong strategic alliances with other companies (food or otherwise) to reduce costs of shipping and logistics; for instance, using another company's logistics provision in order to sell one-off products and services that would otherwise be cost-prohibitive for the SME.

SS can create many opportunities for companies both large and small. Many barriers can exist that prevent companies from adopting such technologies. The usual limitations of cost, worker skills, and knowledge are standard impediments that can be dealt with through suitable support mechanisms; however, this is likely to take time to achieve. SS should not be the realm of the larger companies only. SMEs have the opportunity to adopt internet-based and Smart technologies, thus enabling them to continue to operate in this increasingly pressurized environment.

## 6. The Future Development of Smart Systems for Food Manufacturing Companies

The features described above have been explored in depth by the referenced authors. However, it is useful at this point to bring these together in terms of the wider framework of smart system benefits. Therefore, this section of the paper addresses the second objective of the research, that of *proposing a conceptual system architecture for effective SS implementation*. One such approach towards identifying the range of SS technologies that can be applied within companies can be through the "digital compass" [43]. The company shown in Figure 2 aligns with the 8 basic value drivers and 26 practical SS levers. A further analysis of the compass shows that the technologies can be further divided into two sections, namely the 'responsive' drivers and enablers, which can be described broadly as operations based on, and principally delivering, internally focused benefit, and 'proactive' drivers and enablers, which are broadly externally focused on aligning capabilities with customers' needs. These segments of the digital compass align themselves closely with the SS and Sustainability drivers shown in Table 1. As discussed in the Results section, the perceived preference and focus of the companies on process and operational improvements will lead them to the right-hand side of the compass, while companies that need to maintain market agility and responsiveness to new opportunities will be directed towards the left of the compass. Clearly, the two are not mutually exclusive, and management teams will often desire a mix-and-match model; however, this work clearly shows that, by clarifying the future vision, companies can select appropriate segments of smart system utilization rather than be forced into an expensive "across the board" business transformation.

At the outset of this work, it was acknowledged there is no single prescriptive guide or model to direct Food Manufacturing Companies (FMCs) toward an implementation model that will meet their exact requirements in the shortest possible time and with the smallest implementation cost. However, it is an important step in creating any implementation strategy to recognize known standards and system capabilities to deliver the required benefits. In addition, the results of the survey, especially the Stage 2 interviews, reveal that the challenges for the attraction and retention of human resources with the specialist skills required are common to both categories of company. This suggests a similar starting point for smart system implementation strategies, which then diverge according to the relevant mix and match of drivers and implementation levers that are selected from around the digital compass. Fine tuning the model for strategy selection requires further work to dissect the skills and system architecture required according to the levers to be employed; however, this is very achievable in this specific sector, where regulatory requirements and common process steps have driven some level of standard capability (and in turn the feature of isolation that is hindering ongoing development). This phase of analysis also highlights the importance of creating a decision point to drive strategy formulation and focus on both specific benefit and core competencies for effective

smart system implementation. Achieving this focus will help to overcome the industry's perception of implementation costs and skill development.

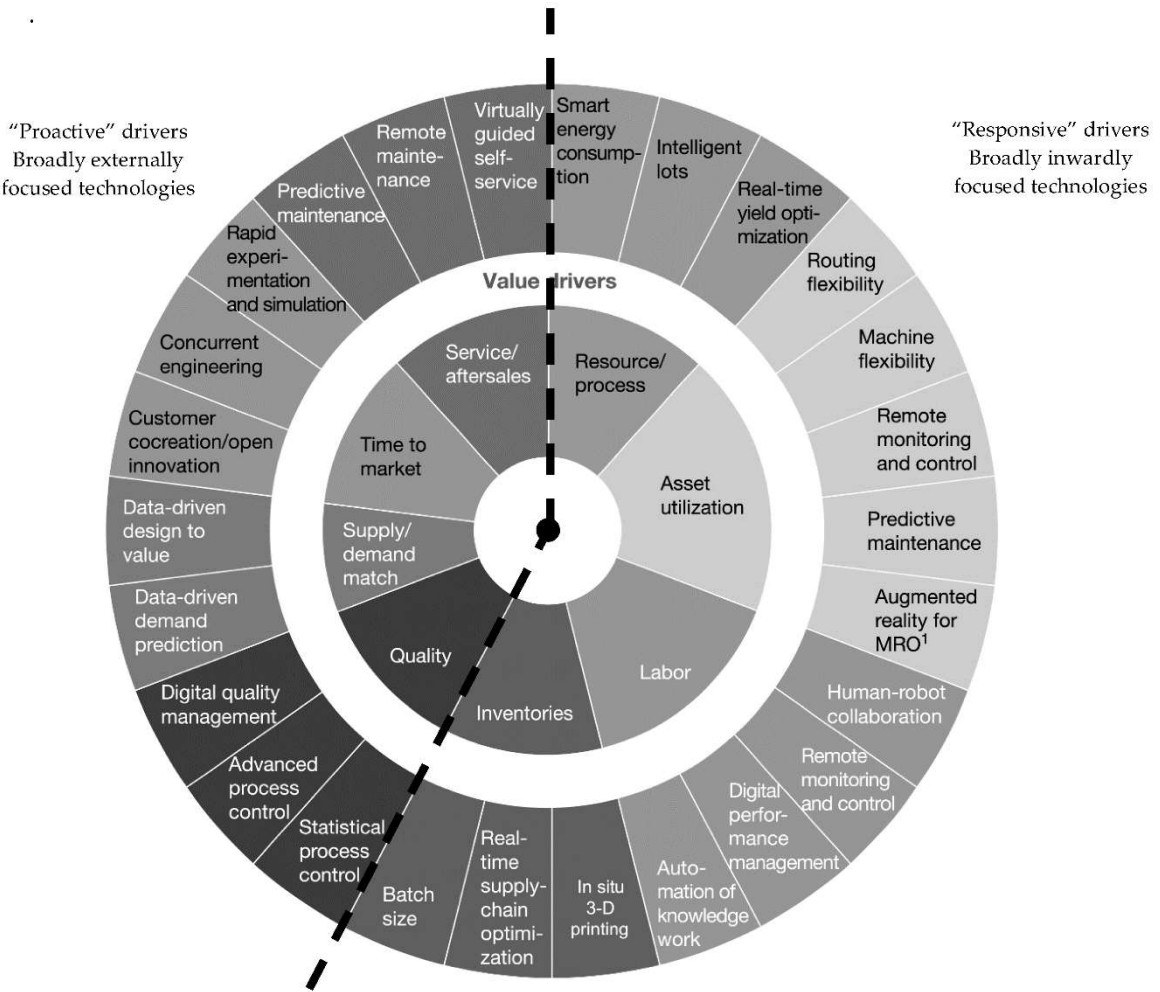

**Figure 2.** Digital compass [43].

*Towards the Implementation of Smart Systems*

Implementing an effective strategy requires an alignment of all of the variables that were explored in this study. This third and final phase to the study recognises that, as in any process, the strategy definition process can be simply shown in terms of the inputs, outputs, and controls that effect the process itself. This paper has considered these variables with a view to identifying the greatest opportunity for food producers to exploit the potential of SS and to link these to their appropriate sustainability dimensions. By engaging with the initial 32 food manufacturing companies, the business drivers were well-articulated and split between internal and external forces. Then, the academic and vocational data sources were examined to understand the key enabling factors, the core supply chain requirements, and the traditional improvement paradigms, such as lean manufacturing, and how they are used to drive traditional productivity gains. Finally, the split between proactive and responsive improvement levers utilized by smart systems has been considered, especially with regard to the projected benefits. Figure 3 shows a schematic of the SS implementation framework. The diagram shows the required inputs into the system. These consist of the enablers, drivers, factors, and capabilities that are needed for the correct implementation of the SS strategy. The resulting outputs of the framework show the proactive and response levers that lead to improved business performance. The following section details the drivers, inputs, and outputs of the framework.

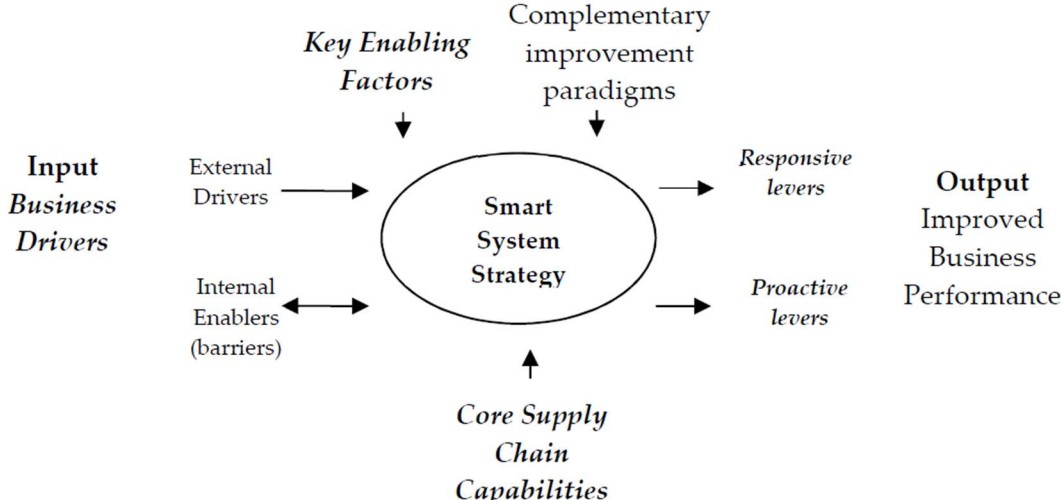

**Figure 3.** A schematic of the smart systems implementation framework.

*Business drivers*: External drivers, including political and environmental factors, changing workforce demographics, and changing customer requirements. Internal dimensions, such as the attraction and retention of staff, training in requisite skills, and system implementation costs.

*Key enabling factors*: Including Big data and knowledge-based automation; (2) Smart Systems; (3) Advanced Robotics; (4) Cloud-based systems; (5) new management paradigms; and (6) new skills and knowledge bases. Lean, Agile, and six-sigma improvement paradigms drive both the improvement culture and affect the human factors.

*Core supply chain capabilities*: virtual enterprises, digital marketing, and virtual supply chain environments focusing upon Information Communication Technologies (ICT) and web technologies by partnering/outsourcing companies [59]. Increase the transparency of operations through to the supply chain in order to achieve greater food security and reliability [60]. Sustainability/resource efficiency: resource and energy efficiency, waste management, and recycling [47]. ICT–Networked business processes. Implementing technologies to share design information and product development information. Cyber security systems and the security of food product and process data to ensure that UK food companies protect their product data [61].

*Proactive/outwardly facing smart system levers*: Innovation tools, marketing tools, and the capability to exploit new opportunities in high-value-added products or niche-market products as a strategy for growth [8]. The ICT capability to share information, particularly design information, throughout a product's life cycle, which will help customers to access this data before any purchase commitments [62]. Open collaboration activities between food companies operating in a trusted and truly collaborative environment will be key to developing and sustaining food manufacturing systems, especially in small food manufacturing companies [61].

*Responsive/inwardly facing smart system levers*: The rapid configuration of food manufacturing systems to be able to ramp up production, or reduce productive capacity where required. This will not only need flexible manufacturing systems, but also flexible working contracts and people. High volume, low variety versus low volume, higher variety will be the likely feature of food producers in the UK [62]. Technology developments for automation, process control, flexible machine control, and enhancing safety aspects in food manufacturing, including new manufacturing technologies, the integration of technologies, and novel structures and techniques [63].

## 7. Conclusions

Food Manufacturing Companies in the UK face many challenges and opportunities to achieve economic sustainability. One such opportunity is through the application and implementation of Smart Systems. This study has attempted to develop an understanding of the attitudes and priorities

of FMCs to the adoption of SS. Through the application of a new measuring tool that was developed in this paper, the research team has been able to profile a range of small-, medium-, and large-scale food manufacturing companies and to determine the strategic drivers and challenges that these companies have in the implementation of SS. Therefore, the initial contribution of this paper is to propose the development of a unique measuring tool for assessing a company's preparedness and its operational and strategic capabilities for the adoption of SS technologies and systems. Through the use of this profiling tool and the adoption of the two-stage research approach, the research team has been able to identify a complex range of company demands and pressures, which indicates that a one-size-fits-all strategy for supporting such companies is going to be largely ineffective and costly [46].

From a theoretical viewpoint, this study contributes to the emerging literature on the relationship between food companies and their motivations for implementing SS and its connection to the dimensions of sustainable production by contrasting the effect of the external and internal pressures and drivers in FMCs [46]. More specifically, the work provides for a more qualitative understanding and clarification with regard to opportunities and challenges that are considered to be relevant for SS implementation and value creation within the food production industry.

In this study, the issue of a company's preparedness for SS was examined based on both external and internal drivers. The study showed that external drivers are currently more important than internal drivers in moving towards the implementation of SS in these food manufacturing companies. The external drivers, such as future political changes and the associated potential loss of a low-cost labour workforce, is driving larger food manufacturing companies towards the implementation of responsive Smart Systems. The smaller food producers are focussed on more proactive tools, including how SS can successfully be used to improve efficiencies in small batch manufacturing, time to market, and promotion of the company on a much wider scale than it currently does. Interestingly, companies see that these external drivers outweigh the internal issues, such as training and costs, and seem to be more willing to overcome these internal barriers as the external drivers seem to be greater than the internal resistance that has previously been seen. Furthermore, a simultaneous approach to the issue of implementing Smart technologies in the UK food sector regarding internal and external drivers is another feature of this study, because, in most of the previous studies, the issue of Smart technology implementation is studied from the internal perspective (training, costs, etc. as being barriers towards implementation). Dividing these drivers into internal and external drivers was the main characteristic of this study that led to different results.

The major limitation of this study is the limited sample size obtained for the Stage 1 survey and the Stage 2 interviews. Whilst the total response level of 32 companies enabled the research team to identify a number of key themes around Smart Systems within the food manufacturing industry, the work cannot be considered to have any statistical significance and, therefore, the outputs of the study are to be considered with this limitation in mind. A more comprehensive survey is now underway, and the outputs of the study should provide additional contextual information for the findings shown in this paper due to its increased sample size. Furthermore, the limitations that were found in the outputs of this study have initiated a further and more detailed survey questionnaire, and semi-structured interview programme, for the next phase of this under-researched area.

**Author Contributions:** Conceptualization, A.T. and C.H.-T.; methodology, R.B. and M.F.; validation, R.M.-J. and P.B.; formal analysis, A.T.; investigation, P.B.; resources, R.M.-J.; data curation, M.F.; writing (original draft preparation), C.H.-T.; writing (review and editing), A.T.; visualization, C.H.-T.

**Funding:** No funding was received for this research.

**Acknowledgments:** Dedicated to the memory of Richard Barton.

**Conflicts of Interest:** The authors declare no conflict of interest.

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
