# Peer review of "Smart Systems Implementation in UK Food Manufacturing Companies: A Sustainability Perspective"

_sustainability, doi:10.3390/su10124693_

Round 1

Reviewer 1 Report

The paper is of interesting and undoubtedly up-to date topic and focuses on the problem of purposefulness of implementing Smart Systems as a way of building sustainable processes in the British food industry. Unfortunately both the research and the paper are rather chaotic. In fact when analyze the text of the paper it is difficult to find the relationship between provided research, smart systems implementation and the sustainability factor. Moreover it is difficult to say what “strong” results have you obtained from your research and what is their value when talking about practical implementation. Therefore – in my opinion – the paper can be published but after deep reconstruction of the paper. When improve your paper, please take into account the following remarks:

1)      It is not necessary to introduce section 1.1 while it is the only section in point 1.

2)      Although you describe Smart Systems (section 1.1) – this description is very general and not organized. In fact in this section you write more about food industry than give the description of smart manufacturing systems.

3)      The literature review (section 2) is not a literature review but the description of the food industry problems again. In fact in this section you have to show what research in this field have been done so far and what results have be obtained. In my opinion the first two sections should be shortened to not exceed 3 pages (remember that your results are the most important part of the paper).

4)      When writing about problems of food producing companies you have to describe them more precisely and show how this problems are connected with the sustainability of production processes. I also suggest to make some review in the area of different manufacturing fields connected with the improvements of technological processes which decide about their sustainability such as: human factor (see eg.: Jasiulewicz-Kaczmarek M., Saniuk A., 2015, Human factor in Sustainable Manufacturing, Editors:M. Antona,  C. Stephanidis , Universal Access in Human-Computer Interaction. Access to the Human Environment and Culture,  LNCS Vol. 9178 , pp.444 – 455, DOI 10.1007/978-3-319-20687-5_43),  intelligent and dynamic scheduling (see eg.: Sobaszek Ł., Gola A., Świć A., Predictive scheduling as a part of intelligent job scheduling system, Advances in Intelligent Systems and Computing, Vol. 637, 2018, pp. 358-367, DOI: 10.1007/978-3-319-64465-3_35) or intelligent control of internal transport (see eg: Gola A., Kłosowski G., Application of fuzzy logic and genetic algorithms in automated works transport organization, Advances in Intelligent Systems and Computing, Vol. 620, 2018, pp. 29-36, DOI: 10.1007/978-3-319-62410-5_4). Please remember that all these (also other) factors decide about “smartness” of a manufacturing system (moreover they can be treated as a factors which you analyze in your research).

5)      In general – please try to be concrete in your paper (it is not a novel). In some parts of your paper there is a lot of text but very little data.

6)      Please notice that the figures 2,3 and 4 are not clear (the text is cut).

7)      The future directions for the industry (section 5) are very general and not directly connected with results presented in previous sections. What is the goal of this session?

8)      The presented conclusions are too general and some of them are not directly based on the realized research.

To summarize – the paper can be interested for readers. However it must be improved. Please try to rewrite it taking into account listed above remarks.

Author Response

Thank you for your detailed review of our paper and for allowing us to revise and resubmit to the Journal. Thank you also for the time you have taken to provide feedback and suggestions on how we could improve our submission. 

We have fully revised the paper in line with your review. Due to the significant changes that have been undertaken, we have provided a table of responses to your review, please find attached the work that we have undertaken.

I hope that we have revised the publication sufficiently for consideration for publication. However, please do not hesitate to contact ourselves if you require further updates and information

Thank You

Reviewer 2 Report

Review

First, thanks to the authors, as a result of inviting to this review, the work is interesting and then I will make some suggestions with the purpose of improving the final result. The main strength of this work is its originality. Smart System is a hot topic today and more research should be developed.

1. Intro

Perhaps it would be interesting to further detail what has already been studied, including some references of the Smart System literature, detail the gap that is intended to cover, the objective of the work with its research questions. Right now it is a bit confusing, the objectives are too broad and difficult to answer in the paper.

I think the introduction section should end on line 76, to move section 1.1. to point 2, integrating it with the review of the literature in an epigraph.

Literature review

Apart from the previous comment, the section does not have many bibliographic references to be a revision. You have to include some more quotes. We need to deepen the idea of sustainability and thus strengthen the link with this Journal. For this, the article by Nuñez-Cacho et al. 2018. We should also add some other references on the subject.

3. It would be interesting to provide the technical data sheet of the bibliographic review.

The size of the sample is very low for inference about the food sector in general. However, due to the novelty of the study, the results can be interesting. It would be necessary to justify this aspect clearly, perhaps highlighting it as a limitation at the end, justifying with some reference the validity of the study and perhaps limiting it to small and medium-sized companies as it seems that the composition of the sample is.

It would be good to explain in this section all the methods used. Phase 1 is developed, the results of phase 1 appear, then phase 2 .... It is somewhat messy, first the methods united and then the results united as well.

Bar charts are more suitable for reporting than for a journal, so it would be better to introduce only those that provide valuable information.

4. Results

Once the comments of the resumed ones in the specified order are finished, you must include an additional section "Discussion and Conclusion" where the contributions of the work and contributions are commented in a synthesized way, the degree to which the objectives have been answered and all that interesting deductions from the results obtained. Some of this information is included in section 5. Future direction. This section should be cut and maybe add the limitations.

Author Response

(The authors gave the same response as above.)

Round 2

Reviewer 1 Report

The paper has been improved carefully and all my suggestions and remarks have been taken into account. I think that the paper can be published at present form.

Reviewer 2 Report

To reinforce the sustaniability aspects of the manuscript, it could be useful use references such as  “Family Businesses Transitioning to a Circular Economy Model: The Case of “Mercadona”. Sustainability 201810 that link food sector with sustainability.